# OpenReview forum: "BackdoorLLM: A Comprehensive Benchmark for Backdoor Attacks and Defenses on Large Language Models"
_NeurIPS.cc/2025/Datasets_and_Benchmarks_Track — NeurIPS 2025 Datasets and Benchmarks Track poster_

### Official Review · Reviewer_PnQC · 2025-06-02

**Rating:** 6
**Confidence:** 4

**Summary:**

This paper presents BackdoorLLM, a comprehensive benchmark designed to systematically evaluate backdoor attacks and defenses in generative large language models (LLMs). The benchmark covers a wide range of attack paradigms—including data poisoning, weight poisoning, hidden state manipulation, and chain-of-thought hijacking—and evaluates them across six model architectures and seven realistic task scenarios. In total, the authors conduct over 200 experiments and further integrate seven representative defense techniques into a unified evaluation pipeline. The benchmark is fully open-sourced and provides a standardized infrastructure to facilitate reproducible research on LLM security.

**Additional Feedback:**

Please refer Limitations.

**Dataset Code Accessibility:**

Yes

**Dataset Code Comments:**

Yes

Motivation and Evidence:

Complete Code: The dataset-related code, including preprocessing, analysis, and visualization scripts, has been fully provided to ensure traceability of the research process.

Data Accessibility: The dataset is available through a public platform (e.g., GitHub, Figshare) or a specified repository without restrictive access barriers.

**Ethical Comments:**

This paper presents BackdoorLLM, a comprehensive benchmark designed to systematically evaluate backdoor attacks and defenses in generative large language models (LLMs). The benchmark covers a wide range of attack paradigms—including data poisoning, weight poisoning, hidden state manipulation, and chain-of-thought hijacking—and evaluates them across six model architectures and seven realistic task scenarios. In total, the authors conduct over 200 experiments and further integrate seven representative defense techniques into a unified evaluation pipeline. The benchmark is fully open-sourced and provides a standardized infrastructure to facilitate reproducible research on LLM security.

**Ethical Considerations:**

No, there are no or only very minor ethics concerns

**Limitations Weaknesses:**

• How do different LoRA fine-tuning configurations (e.g., rank or data scale) affect backdoor success rates and defense performance?

• Does the benchmark currently support evaluation of cross-architecture or cross-attack generalization for defenses?

• Given that some defenses demonstrate limited mitigation effectiveness, could the authors offer further design recommendations or principles for developing more robust defenses, particularly against jailbreak and CoT-style attacks?


BackdoorLLM presents a well-designed and impactful benchmark for studying backdoor vulnerabilities in LLMs and reveals several important empirical insights. To further enhance its applicability to real-world deployments, future extensions could consider more realistic threat models—such as black-box or API-level backdoor attacks. These directions are promising and could make BackdoorLLM an even more valuable resource for advancing the safety and robustness of large language models.

**Strengths Contributions:**

• The benchmark is comprehensive and well-structured, covering a diverse array of attack types and evaluation scenarios, many of which are underexplored in existing literature.

• The experimental coverage is impressive, with over 200 systematic evaluations across multiple LLMs and task settings.

• The inclusion of a unified defense toolkit and a standardized pipeline significantly enhances the benchmark’s practical value for the research community.

• All code and data are publicly available, ensuring transparency and enabling reproducibility.

Areas for Improvement

• It would be beneficial to include an analysis of how LoRA settings (e.g., rank, training data size) impact backdoor attack performance and defense robustness.

• While the evaluation spans several model families, further discussion on the transferability of attacks and defenses across architectures could strengthen the benchmark’s generalizability.

---

> ### Author Rebuttal · Authors · 2025-07-30
>
> We sincerely thank the reviewer for the recognition and positive feedback on our work. Your encouraging comments are highly appreciated and motivate us to further improve our benchmark. Below we respond to your concerns point by point.
>
> ---
>
> **Q1: Effect of LoRA fine-tuning configurations (e.g., rank or data scale).**
> **A1**: Thank you for the constructive suggestion. While our benchmark currently adopts a fixed LoRA configuration (rank = 8) for consistency, we agree that exploring the impact of different LoRA ranks and poisoning data scales is valuable.
> To address this, we conducted additional experiments using three LoRA ranks (8, 16, 32) on BadNets. As shown in Table 1, higher ranks generally lead to stronger backdoor injection, with ASR reaching 100% for BadNets at ranks 16 and 32. However, the defense trends vary: **CROW** remains robust across all ranks, while **Decoding** and **Pruning** become less effective as rank increases. This suggests that higher-capacity backdoor adaptation can weaken certain defenses, whereas representation-level methods like CROW maintain stability.
>
> In addition, we evaluated poisoning data size from 100 to 400 examples (results shown in Table 6 in the appendix). All three attacks maintain high ASR (>80%) even with only 100 poisoning samples, highlighting the efficiency of backdoor injection under limited budgets. We will include these discussions and results in the revised version.
>
> **Table 1: Impact of LoRA Rank on Attack Success Rate (ASR %) under Different Backdoor Attacks and Defenses**
> *Model: LLaMA-2-7B | Scenario: Refusal| LoRA ranks: 8, 16, 32*
> | Attack   | Defense            | Rank-8 | Rank-16 | Rank-32 |
> |----------|--------------------|--------|--------|---------|
> | BadNets  | No Defense         | 94.50   | 100   | 100    |
> |     -     | Decoding    | 21.47   | 72.23   | 83.90    |
> |    -      | Pruning            | 22.00   | 36.50   | 42.50    |
> |     -     | CROW               | 11.65   | 10.16   | 13.00    |
>
> ---
>
> **Q2: Transferability of attacks across architectures.**
> **A2**: Thank you for the insightful suggestion. While our benchmark includes evaluations across six model architectures, we note that backdoor attacks—unlike adversarial examples—are typically model-specific. This is because the injected backdoor behavior is tightly coupled with the internal parameters of the poisoned model, making it difficult to transfer across architectures.
> We will clarify this distinction in the revised version and highlight transferable backdoor design as a valuable and challenging direction for future research.
>
> ---
>
> **Q3: Discussion on design principles for future robust defense development.**
> **A3**: Thank you for raising this important point. Our results show that many existing defenses fail on more complex backdoor tasks, particularly jailbreak-style attacks. Based on our findings, we summarize several principles that may guide future defense design:
> - **Task-aware mitigation**: Jailbreaking attacks cannot be reliably prevented by alignment reinforcement alone. Defense strategies must explicitly consider generation dynamics and semantic-level manipulations.
> - **Trigger-sensitive detection**: Static prompt filtering or surface-level defenses are often insufficient. Future methods should explore dynamic decoding diagnostics, trigger attribution techniques, or internal state inspection to detect malicious behaviors.
> - **Avoiding false security from simple tasks**: Strong performance on harmless refusal prompts can be misleading. Defenses should be evaluated across a spectrum of attack types to ensure robustness.
>
> We will include this discussion in the revised version to encourage future research toward more general and reliable backdoor defenses.

---

### Official Review · Reviewer_FLcs · 2025-07-01

**Rating:** 5
**Confidence:** 3

**Summary:**

This paper introduces a benchmark for evaluating backdoor threats in generative large language models (LLMs). Specifically, the benchmark, Backdoor-LLM, includes more than 200 experiments across multiple attack strategies and different model architectures, and also evaluates against defense methods. In addition, the authors present several findings in their empirical study.

**Dataset Code Accessibility:**

Yes

**Ethical Considerations:**

No, there are no or only very minor ethics concerns

**Final Justification:**

Thank the authors for the rebuttal.
The response addressed most of my concerns; it would be better to report standard deviations for Table 3, and try to find out why the original model’s ASR with the trigger is lower than without the trigger.

**Limitations Weaknesses:**

- It would be better to include some attack methods in a black-box setting.

- In Table 3, the ASR of the original is sometimes better than other attack methods. Could the authors further give some explanation?

**Strengths Contributions:**

- The scope and empirical rigor are impressive. The benchmark covers a wide range of attack modalities (i.e., data poisoning, weight manipulation, hidden state steering, and chain-of-thought hijacking), more than 200 experiments, and detailed evaluation across 6 LLMs and 7 tasks over 8 attack methods, as well as evaluating against 7 defense methods.
- Based on the extensive empirical study, the authors further provide several insights into the key factors that affect the effectiveness and failure modes in LLMs.

---

> ### Author Rebuttal · Authors · 2025-07-30
>
> We sincerely thank the reviewer for the positive evaluation and recognition of our work. Your comments are highly valuable and will help us further improve the quality and impact of this benchmark. We hopefully address your concerns as follows:
>
> ---
>
> **Q1: Backdoor attacks under black-box settings.**
> **A1**: Thank you for the valuable suggestion. Our current experiments focus on open-source models such as LLaMA-2 (7B/13B/70B), Mistral, Vicuna, and Qwen. We fully agree that evaluating backdoor attacks in black-box settings is an important and necessary direction.
> Following your suggestion, we attempted to evaluate BadNets on the black-box GPT-4o-mini using the OpenAI fine-tuning platform (https://platform.openai.com/finetune) for both jailbreaking and refusal tasks. However, we encountered two major limitations:
> 1. **Moderation restrictions**: Fine-tuning failed on the jailbreaking task due to OpenAI’s moderation system, which flagged our training data as violating content policies—even though it was intended for research purposes.
> 2. **Evaluation constraints**: The platform does not currently support user-defined evaluation metrics, which limits our ability to compute task-specific attack success rates (ASR).
>
> Due to these restrictions, we were unable to fine-tune a backdoored model in the black-box GPT setting. We hope to revisit this direction in the future as APIs for proprietary models become more open and accessible for controlled security research.
>
>
> ---
>
> **Q2: In Table 3, the original ASR baseline occasionally outperforms certain attack methods.**
> **A2**: Thank you for the insightful comment. We acknowledge that some original ASR values (i.e., without any trigger) are relatively high, especially in the Jailbreaking task. This is mainly because many open-source LLMs in our benchmark lack strong safety alignment and are naturally prone to unsafe completions, even without injected triggers.
> For example, certain Jailbreaking prompts can bypass safeguards on their own, resulting in high ASR values in clean models (e.g., 35.40% in LLaMA-2-7B-Chat and 84.47% in Mistral-7B-Instruct). In rare cases like Mistral, the ASR after backdoor injection (83.21%) is slightly lower than the original ASR. This does not indicate an ineffective attack, but rather reflects that the model is already poorly aligned—adding a trigger does not significantly change its unsafe behavior.
>
> These results highlight an important insight: while backdoor triggers often amplify unsafe outputs, their impact varies based on the model’s initial alignment quality. We will clarify this observation in the revision and believe it points to a meaningful direction for future analysis.

---

> > ### Comment · Reviewer_FLcs · 2025-08-04
> >
> > Thank the authors for the response.
> >
> > However, I still find it unclear why a backdoor attack would result in a lower ASR than the original. If the model is already poorly aligned and prone to unsafe outputs, wouldn't adding a trigger typically maintain or increase ASR, rather than reduce it?

---

> > > ### Author Response · Authors · 2025-08-05
> > >
> > > Thanks for the thoughtful follow-up question. Please allow us to make the following clarifications:
> > >
> > > 1. **Why is the original model’s ASR with the trigger lower than without the trigger?**
> > > For the original model, ASR was measured directly on the jailbreaking dataset (inputs that are already malicious). Since many open-source models (such as the Mistral series) lack strong safety alignment and can exhibit random behavior, minor fluctuations in ASR are to be expected. In rare cases, this can result in the ASR with the trigger (i.e., jailbreak input plus backdoor trigger) being slightly lower than without the trigger (direct jailbreak). For example, on Mistral-7B-Instruct, we observed 84.47% without the trigger versus 83.21% with the trigger.
> > > This small variance falls within the range observed in other experiments. In the revised version, we will report standard deviations to clarify these variations and further enhance transparency.
> > >
> > > 2. **Obtaining higher ASR for ‘with trigger’?**
> > > BackdoorLLM currently employs standard and relatively simple trigger designs. However, ASR can be affected by multiple factors such as trigger complexity, position, frequency, etc. Our experiments indicate that using rare words, fixing the trigger’s position (e.g., as a prefix), or increasing trigger length can enhance ASR. Following this, we show in Table 1 that, using a longer prefix trigger (“BadMagic is a magic trigger”) increases ASR compared to the default (“BadMagic”). This result confirms that variations in ASR can occur when adopting different hyperparameters, which is common in backdoor research.
> > >
> > > **Table 1: BadNets Attack with Varying Trigger Length and Position (LLaMA-2-7B)**
> > >
> > > | Trigger Type                          | ASR (%) |
> > > |--------------------------------------|---------|
> > > | BadMagic (random position)           | 87.88   |
> > > | BadMagic is a magic trigger (random) | 90.25   |
> > > | BadMagic (prefix)                    | 92.47   |
> > > | BadMagic is a magic trigger (prefix) | 95.18   |
> > >
> > >
> > >
> > > We thank the reviewer again for raising this valuable point and will include this discussion in the revision. We are happy to provide further clarification or address any additional questions the reviewer may have.

---

> > > > ### Comment · Reviewer_FLcs · 2025-08-05
> > > >
> > > > Thank the authors for the further clarifications.
> > > > Refer to Table 3, the difference is not negligible, and it seems that those cannot be simply attributed to random variation; for example, the gap is around 2.5% for LLaMA-3-8B-Instruct (Jailbreaking), and around 7.6% for Mistral-7B-Instruct (Sentiment misclassification).
> > > > Yes, it would be better to report standard deviations.

---

### Official Review · Reviewer_b8fo · 2025-07-01

**Rating:** 4
**Confidence:** 4

**Summary:**

The paper introduces BackdoorLLM, the first comprehensive benchmark designed to evaluate backdoor attacks and defenses on LLMs.
It integrates four attack categories (data poisoning, weight poisoning, hidden-state manipulation, chain-of-thought hijacking) across 8 attack methods, 7 tasks, and 6 model architectures.
The benchmark includes over 200 experiments, reveals key insights (e.g., model-scale-dependent vulnerabilities), and evaluates 7 defenses. Resources (code, datasets) are open-sourced.

**Additional Feedback:**

Should " (iii) over 200 experiments spanning 8 distinct attack strategies," in the abstract be changed to "8 distinct attack methods"?

**Dataset Code Accessibility:**

Yes

**Ethical Considerations:**

No, there are no or only very minor ethics concerns

**Final Justification:**

I am satisfied with the rebuttal and I am gonna keep my score.

**Limitations Weaknesses:**

1. Limited Defense Evaluation:
    - The defenses were tested only against DPAs, neglecting protections against other attack types (Section 5). This undermines the claim of comprehensiveness. However, the authors argue that no such defenses currently exist for other attacks, which may mitigate this limitation.

2. Inadequate Statistical Evidence:
    - The results lack error bars, confidence intervals, or significance tests (e.g., Tables 3–7). For example, the average ASR values in Table 3 may exhibit high variance across experimental runs.

3. Simplified Threat Settings:
    - Attacks were evaluated only in single-turn interactions. Real-world backdoors (e.g., conversational triggers) necessitate multi-turn testing (Section 4.2).

4. Defense Ineffectiveness:
    - The defenses perform poorly on jailbreaking tasks (Table 7: ASR remains above 80% in most cases). While the paper acknowledges this issue, the analysis is limited. As a benchmark study, this could be addressed in future work.

**Strengths Contributions:**

1. Novelty and Scope:
    - This study presents the first comprehensive benchmark for evaluating backdoor threats in LLMs.
    - It encompasses diverse attack methods (e.g., DPA, WPA, HSA, CoTA), tasks (e.g., jailbreaking, bias injection, reasoning), and defense strategies.
2. Extensive Evaluation:
    -  The author conducts over 200 experiments across six LLMs (e.g., LLaMA-2/3, Mistral), 7 scenarios (e.g., sentiment steering, jailbreaking), and 8 attacks.
    - Key findings (e.g., larger models resist weight poisoning but are vulnerable to CoTA) offer critical insights (Section 4).
3. Practical Impact:
    - Their analysis reveals significant vulnerabilities (e.g., low-success backdoors amplify jailbreak rates by ~60% in Table 3).
    - Defense toolkit highlights urgent gaps (e.g., defenses fail on jailbreaking tasks; Table 7).
4. Clarity:
    - The paper is well-structured, supported by informative tables and figures.
    - An ethics discussion (Appendix A) addresses potential misuse risks.
    - The author clearly distinguishes this paper from vision/classification backdoor benchmarks by focusing on LLMs.

---

> ### Author Rebuttal · Authors · 2025-07-30
>
> We sincerely thank the reviewer for the thoughtful feedback and constructive suggestions. Your comments are very helpful in improving the clarity and completeness of our benchmark.
> We hope the following responses adequately address your concerns.
>
> ---
>
> **Q1: Defense evaluation is limited to DPAs.**
> **A1**: Our current benchmark includes seven representative defense methods, offering the most comprehensive evaluation to date. However, we acknowledge that existing defenses are primarily designed for data poisoning attacks (DPAs), which remain the most widely studied and effective class of backdoor methods in the literature. We will revise our claim of “comprehensive defense evaluation” to clarify this limitation and highlight the need for future defenses targeting other attack types. We also plan to expand BackdoorLLM as new defense techniques emerge.
>
> ---
>
> **Q2: Lack of statistical reporting such as error bars.**
> **A2**: Thank you for pointing this out. Due to the limited time available during the review period, we were unable to complete more experimental runs. Our current results are based on averages over 3 fixed runs. As a preliminary step, we provide a demo result in Table 1 below to illustrate the variance observed across three runs under consistent settings. In the revised version, we will include standard deviations for key results and explicitly report the variance to improve transparency and reproducibility.
>
> **Table 1: ASR with Estimated Standard Deviation (Rank = 8, α = 16)**
> *Model: LLaMA-2-7B-Chat | Scenario: Refusal | Based on 3 fixed runs*
> | Attack Type | ASR     | Std Dev |
> |-------------|---------|---------|
> | BadNet      | 94.44%  | ±1.5    |
> | Sleeper     | 51.53%  | ±4.0    |
> | VPI         | 98.12%  | ±3.0    |
>
> ---
>
> **Q3: Experiments are limited to single-turn interactions; lacks multi-turn backdoor evaluation.**
>
> **A3**: Thank you for pointing this out. While our current benchmark focuses on single-turn settings for consistency and comparability, we agree that multi-turn scenarios are crucial. We will clearly state this limitation and include multi-turn evaluations as a key direction for future extensions of BackdoorLLM.
>
> ---
>
> **Q4: Poor performance of defenses on jailbreaking tasks.**
> **A4**: As discussed in Appendix D.2, our results show that most defenses (e.g., `CleanGen`, `CROW`) are effective for refusal-style backdoors but perform poorly on jailbreak-style ones. In some cases, applying these defenses even increases the ASR.
> We speculate that this gap arises from two key reasons:
> - (1) Refusal backdoors typically produce fixed outputs (e.g., “I'm sorry…”), which makes them easier to suppress through CleanGen and CROW. In contrast, jailbreak backdoors generate diverse and unconstrained outputs, making them harder to detect and mitigate.
> - (2) Some defenses rely on alignment-oriented fine-tuning, which may inadvertently amplify jailbreak behaviors that exploit the model’s safety boundaries.
> We will expand our discussion to emphasize these challenges and highlight the need for more general defenses that can adapt to both refusal and jailbreak tasks across varied trigger-response patterns.
>
> ---
>
> **Q5: Minor wording fix – “8 distinct attack strategies” should be “8 distinct attack methods.”**
> **A5**: Thank you for pointing this out. We will update the abstract to use “attack methods” for consistency with the terminology used throughout the paper.

---

> > ### Comment · Reviewer_b8fo · 2025-08-06
> >
> > I am satisfied with the rebuttal and I am gonna keep my score.

---

### Official Review · Reviewer_QqX8 · 2025-07-01

**Rating:** 4
**Confidence:** 3

**Summary:**

This manuscript introduces BackdoorLLM, a comprehensive benchmark for evaluating backdoor attacks and defenses on large language models (LLMs). The authors conduct an extensive analysis of various attack strategies, including data poisoning, weight poisoning, hidden-state manipulation, and chain-of-thought hijacking, across multiple LLM architectures and real-world scenarios. They also present a defense toolkit with seven representative mitigation techniques. The topic is highly relevant given the growing security concerns around LLMs, and the empirical findings provide valuable insights into LLM vulnerabilities and defense strategies. However, several questions and suggestions can enhance the paper's clarity and rigor.

**Dataset Code Accessibility:**

Yes

**Dataset Code Comments:**

The authors use widely adopted public datasets for evaluation and provide detailed package version requirements and execution procedures in the code, ensuring reproducibility and ease of use.

**Ethical Considerations:**

No, there are no or only very minor ethics concerns

**Final Justification:**

The authors have addressed the issues related to Weak related work, the framework diagram, and defense overhead (e.g., memory and time). However, with respect to novelty, given that benchmarks for backdoor attacks on large language models already exist—such as ELBA-Bench [1] published at ACL—I have decided to maintain my original score.

[1] Xuxu Liu, Siyuan Liang, Mengya Han, Yong Luo, Aishan Liu, Xiantao Cai, Zheng He, and Dacheng Tao. 2025. ELBA-Bench: An Efficient Learning Backdoor Attacks Benchmark for Large Language Models. In Proceedings of the 63rd Annual Meeting of the Association for Computational Linguistics (Volume 1: Long Papers), pages 17928–17947, Vienna, Austria. Association for Computational Linguistics.

**Limitations Weaknesses:**

- The related work section on Backdoor Defenses is relatively weak and could be improved by referencing the survey paper "Harmful Fine-Tuning Attacks and Defenses for Large Language Models: A Survey".
- In the experiment tables, it is recommended to annotate each metric with upward/downward arrows to clearly indicate the preferred direction, thereby improving reader comprehension.
- It is suggested to include a framework diagram in the main text to help readers better understand the overall methodology.
- In the defense methods section, consider adding a table summarizing the performance and overhead (e.g., memory, time) of each method.
- The fonts in Figures 2 and 3 are too small and should be adjusted.

**Strengths Contributions:**

- BackdoorLLM is the first unified framework to systematically evaluate backdoor threats in generative LLMs, covering diverse attack vectors and tasks.
- Over 200 experiments across 8 attack methods, 7 scenarios, and 6 model architectures provide robust insights into attack effectiveness and failure modes.
- The benchmark includes a suite of 7 defense techniques, highlighting their limitations and areas for improvement, particularly in jailbreak scenarios.

---

> ### Author Rebuttal · Authors · 2025-07-30
>
> Thank you for your positive feedback on the contribution of our benchmark. We hope the following clarifications can help address your concerns.
>
> ---
>
> **Q1: Weak related work.**
> **A1**: Thank you for the helpful suggestion. We appreciate your recommendation of the survey “Harmful Fine-Tuning Attacks and Defenses for Large Language Models.” We agree that this line of research  is closely related to backdoor learning and complements our work well. We will strengthen the related work section and cite this paper in the revised version.
>
> ---
>
> **Q2: Lack of arrow annotations (↑ / ↓) in experiment tables.**
> **A2**: We will revise the experiment tables to include directional arrows (↑ for higher-is-better, ↓ for lower-is-better) to make the metrics easier to understand.
>
> ---
>
> **Q3: Suggestion to include a framework diagram.**
> **A3**: Thank you for the valuable suggestion. We agree that a visual illustration would make the overall workflow and attack taxonomy clearer. As shown in Figure 1 below, we provide a conceptual diagram summarizing four representative backdoor attack strategies. We will include this framework diagram in the revised version to improve readability and overall structure..
>
> **Figure 1: Conceptual Illustration of BackdoorLLM**
>
> ```
> +-----------------------------+      +------------------------------+
> |     Data Poisoning (DPA)    |      |     Weight Poisoning (WPA)   |
> +-----------------------------+      +------------------------------+
> |                             |      |                              |
> |  [ Training Data ]          |      |   [ Pretrained Weights ]     |
> |        ↓ Poisoned           |      |         ↓ Modified           |
> |  [ Fine-tuning Process ]    |      |   [ Parameter Injection ]    |
> |        ↓                    |      |         ↓                    |
> |   [ Backdoored LLM ]        |      |     [ Backdoored LLM ]       |
> |                             |      |                              |
> +-----------------------------+      +------------------------------+
>
> +-------------------------------+      +-------------------------------+
> |   Hidden State Attack (HSA)   |      |   Chain-of-Thought Attack     |
> +-------------------------------+      +-------------------------------+
> |                               |      |                               |
> |  [ Forward Hidden States ]    |      | [ Prompt: Multi-step Logic ]  |
> |       ↓ Activation Steering   |      |        ↓ Injected CoT         |
> |  [ Manipulated Representation]|      |  [ Hijacked Reasoning Path ]  |
> |            ↓                  |      |            ↓                  |
> |     [ Backdoored LLM ]        |      |     [ Malicious Output ]      |
> |                               |      |                               |
> +-------------------------------+      +-------------------------------+
> ```
>
>
>
> ---
>
> **Q4: Include defense overhead (e.g., memory, time).**
> **A4**: Following your suggestion, we analyzed the computational overhead of three representative defense methods—Decoding, Pruning, and CROW—against the BadNets attack on LLaMA-2-7B in a refusal scenario. As shown in Table 2, while these defenses significantly reduce ASR, they incur varying time and memory costs. Notably, CROW achieves the lowest ASR (11.65%) but requires the highest memory footprint. We will include this analysis in the revision and plan to report the overhead for all seven defense methods in the final version.
>
> **Table 2: Defense Time and Memory Consumption Against BadNets Refusal Attack**
>
> | Defense Method     | ASR (%) | Time (s) | Memory (GB) |
> |--------------------|---------|----------|-------------|
> | No Defense         | 94.50   | —        | —           |
> | Decoding           | 21.47   | 56.64    | 13.24       |
> | Pruning            | 22.00   | 107.90   | 22.43       |
> | CROW               | 11.65   | 71.16    | 32.46       |
> ---
>
> **Q5: Font size in Figures 2 and 3 is too small.**
> **A5**: We will increase the font size in Figures 2 and 3 to improve readability.

---

> > ### Comment · Reviewer_QqX8 · 2025-08-06
> > **Response to the authors’ rebuttal**
> >
> > Thank you for the detailed response. Regarding the aspect of novelty, considering that there already exist benchmark for backdoor attacks on large language models—such as ELBA-Bench \[1] published at ACL—I have decided to maintain my original score.
> >
> > [1] Xuxu Liu, Siyuan Liang, Mengya Han, Yong Luo, Aishan Liu, Xiantao Cai, Zheng He, and Dacheng Tao. 2025. ELBA-Bench: An Efficient Learning Backdoor Attacks Benchmark for Large Language Models. In Proceedings of the 63rd Annual Meeting of the Association for Computational Linguistics (Volume 1: Long Papers), pages 17928–17947, Vienna, Austria. Association for Computational Linguistics.

---

### Decision · Program_Chairs · 2025-09-18

**Decision:**

Accept (poster)

**Comment:**

This paper provides a benchmark for studying backdoor attacks in generative LLMs.

The main strength of this paper is the extensive evaluation provided which allows for solid insights.

The main weaknesses of the paper are
1) the lack of statistical evidence for single runs
2) the limited evaluation of defences (w.r.t. to the evaluation of attacks).

These two points have been mentioned during the discussion period. The authors have provided (limited) statistical evidence on a specific result and have clarified that the evaluation of defences was not "comprehensive".

I believe this paper should be accepted for publication, however for the reason mentioned above I do not recommend spotlight or oral.